# A New Challenge: Path Planning for Autonomous Truck of Open-Pit Mines in The Last Transport Section

**Ziyu Zhao** and **Lin Bi** *

School of Resources And Safety Engineering, Central South University, Changsha 410083, China; 185512099@csu.edu.cn
* Correspondence: mr.bilin@csu.edu.cn

**Abstract:** During the operation of open-pit mining, the loading position of a haulage truck often changes, bringing a new challenge concerning how to plan an optimal truck transportation path considering the terrain factors. This paper proposes a path planning method based on a high-precision digital map. It contains two parts: (1) constructing a high-precision digital map of the cutting zone and (2) planning the optimal path based on the modified Hybrid A* algorithm. Firstly, we process the high-precision map based on different terrain feature factors to generate the obstacle cost map and surface roughness cost map of the cutting zone. Then, we fuse the two cost maps to generate the final cost map for path planning. Finally, we incorporate the contact cost between tire and ground to improve the node extension and path smoothing part of the Hybrid A* algorithm and further enhance the algorithm's capability of avoiding the roughness. We use real elevation data with different terrain resolutions to perform random tests and the results show that, compared with the path without considering the terrain factors, the total transportation cost of the optimal path is reduced by 10%–20%. Moreover, the methods demonstrate robustness.

**Keywords:** intelligent mining; autonomous haulage; planning map; Hybrid A* algorithm; path planning

---

## 1. Introduction

Cost control has always been the top priority in the mining industry, and how to control the cost is one of the critical points for mining companies to make profits. In the open-pit mines, the transportation cost accounts for about 50% of the total mining cost [1–3]. In the absence of mining technology innovation, how to improve transportation efficiency and reduce extra transportation costs are crucial to improve the profits of mining enterprises. The rise of driverless technology makes the problem have a new solution. At present, unmanned ground vehicles (UGVs) have been widely used in port or warehouse automation logistics, disaster relief, military, and other industrial and military fields [4–8].

In the mining field, autonomous navigation of the load-haul-dump (LHD) machine in underground mines has been successfully applied [9,10]. However, it is still a challenge to apply autonomous technology in open-pit mining. The open-pit mining area is an ideal environment for unmanned vehicles due to its closed environment and few external interference factors. Komatsu and other mining equipment manufacturing giants have carried out unmanned transportation experiments in a Chilean copper mine since 2008 [11]. Their results show that the transportation capacity is increased by 40% and the cost has reduced by 15% in the autonomous haulage system (AHS). Another unmanned transportation experiment in Australian mining also shows that unmanned transportation has more

considerable economic significance compared with traditional transportation [12]. Nonetheless, Ayres [13] and Juliana Parreira [14] both pointed out that the AHS may have problems such as unable to respond to ground pits or rocks promptly, resulting in extra fuel consumption and tire wear. Also, some studies [3,15] have concluded that, when haulage trucks run on poor road conditions, the vehicle cost will be increased. Kansake [16], through the simulation experiment, concluded that under the general mining road conditions, the dynamic load of the tire would be 60% higher than the static load, which will aggravate the tire wear. For the existing AHS, only the large obstacles on the ground are considered in path planning. In contrast, the tiny obstacles are ignored, like rocks or terrain undulation on the surface, resulting in additional tire wear. Thus, the truck route should be planned on smoother ground.

The path planning of UGV is divided into route planning, trajectory planning, and motion planning [17]. Route planning is point-to-point planning, which regards the vehicle as a particle and does not consider other constraints, so it requires less map accuracy. Trajectory planning is based on route planning, with more constraints of vehicles, such Cering the most constraints, such as vehicle control strategy. Choi et al. [18] used the map from Google Earth to optimize the route of dump trucks in open-pit mines. Google Earth only provides a low-resolution surface elevation model, but the realization of trajectory planning requires more accurate surface topographic maps.

The digital surface model (DSM) is a model that contains the surface elevation information, and it is generally used for the reconstruction of the terrain model as its high precision. There are several methods to build a DSM: traditional digital mapping [19], unmanned aerial vehicle (UAV) lidar mapping [20,21], oblique photogrammetry [22], etc. Among these methods, the oblique photogrammetry has been widely used in urban surveying and mapping [23], disaster monitoring [24], and the open-pit mining industry [25–27] because of its high precision and convenient operation. Based on DSM, trajectory planning can be carried out. Jurgen Zoto et al. [28] planned a coverage path for UGV of vineyard based on DSM. Kai Zhang et al. [29] used lidar to measure the surface elevation point cloud and constructed the ground map to carry out the path planning of cross-country vehicles. These DSM-based path planning studies only separated obstacle information but did not consider surface roughness information in the planning map.

The transportation environment of open-pit mines is unstructured, and the Hybrid A* algorithm (HA* algorithm) is a heuristic algorithm widely used in real-time vehicle path planning for unstructured environments [30–34]. This algorithm adds the non-holonomic constraints into the heuristic functions and plans the path in the Voronoi field map. Nevertheless, in the planning process, the regular Hybrid A* algorithm only considers the large obstacles. However, it ignores the little pit or roughness of the surface, which may cause an additional cost for large vehicles.

There are a lot of open spaces in the cutting zone of open-pit mines without any obstacles. In this scenario, the path planner considers the road surface as a passable area without obstacles, thus planning a path directly to the loading position which may result in extra tire cost because of the ground roughness. In order to reduce the extra cost, this paper proposes a high-precision planning map construction method based on DSM and an improved Hybrid A* algorithm. This planning map contains both surface obstacles and roughness information. Firstly, the DSM is divided into an obstacle point set and surface roughness point set with different terrain features. Secondly, the obstacle point set is calculated as obstacle cost map (OCM) according to generalized Voronoi diagram (GVD). Then the sliding window standard deviation method is used to process the surface roughness point set and generate the roughness cost map (RCM) without obstacle. Finally, the two maps are integrated into the cost map (C-MAP) for path planning, which contains the information of obstacles and roughness on the road. With this C-MAP, we modified the Hybrid A* algorithm by adding the contact cost between the tire and ground into the heuristic function, so that the path can be planned on the smooth road surface. The main contributions of this paper are as follows:

1.  We propose a truck path planning method based on high-precision digital map for the final transportation section of open-pit mines, which can effectively reduce the driving cost of trucks.

2.  We design a planning map that contains obstacle information and roughness information on the road surface and its construction method.
3.  We improve the Hybrid A* algorithm to incorporate the roughness information into the cost estimation function of node extension.

The structure of this paper is as follows: Section 2 describes the modified Hybrid A* algorithm in detail, including planning map building method and algorithm improvement. In Section 3, we evaluate the algorithm performance in different real open-pit cutting zone scenes and analyze the algorithm robustness through a thousand random tests with different terrain resolution maps. Conclusions and future work are presented in Sections 4 and 5.

## 2. Materials and Methods

To avoid the rough area on the transportation road, it is necessary to make a planning map containing the roughness information and add it to the estimation function of the Hybrid A* algorithm. The method of constructing a planning map with roughness information is as follows:

1.  DSM data acquisition and generation. The raw elevation data of cutting zone is obtained by some topographic mapping methods (oblique photogrammetry, lidar scanning, etc.), and the corresponding DSM is generated.
2.  Constructing the obstacle cost map of the cutting zone. Firstly, the obstacle point set in DSM is filtered by setting different terrain factors thresholds. Then the obstacle binary map is constructed by the elevation scanning method, and finally, the obstacle cost map is made from GVD.
3.  Constructing the roughness cost map of the cutting zone. Firstly, removing the obstacle point set in DSM, filtering the remaining point set with another terrain threshold. Then, using the standard deviation of moving window to traverse calculation, its value is taken as the roughness value of all coordinate points within the window range. Finally, normalizing the roughness matrix to generate the roughness cost Map of the cutting zone.
4.  Generating the C-MAP. Overlaying and normalizing the value matrix of obstacle cost map and roughness cost map, get the C-MAP, which is used in path planning.

The modified parts of the Hybrid A* algorithm are as follows:

1.  Modify the $g$-value estimation function. The contact cost between the tire and the ground surface is added into the $g$-value estimation function. Consequently, the step length a dynamic value instead of a static value in the part of the node extension.
2.  Modify the conjugate gradient object function. The original Voronoi term is replaced by a new roughness term to correct the path points passing through rough area.

The primary process of map construction and modified Hybrid A* algorithm is shown in Figure 1.

### 2.1. Raw Data Collection and DSM Generation

There are many topographic surveys to collect DSM original image data. In this paper, UAV oblique photogrammetry is used for data collection. After that, using terrain processing software to extract the elevation points from the original image and generating the DSM, as shown in Figure 2. Save the elevation points information of DSM in the following matrix:

$$E = \begin{bmatrix} e_{11} & \cdots & e_{1n} \\ \vdots & \ddots & \vdots \\ e_{m1} & \cdots & e_{mn} \end{bmatrix}, \tag{1}$$

where $e_{ij}$ is the corresponding coordinate of elevation points.

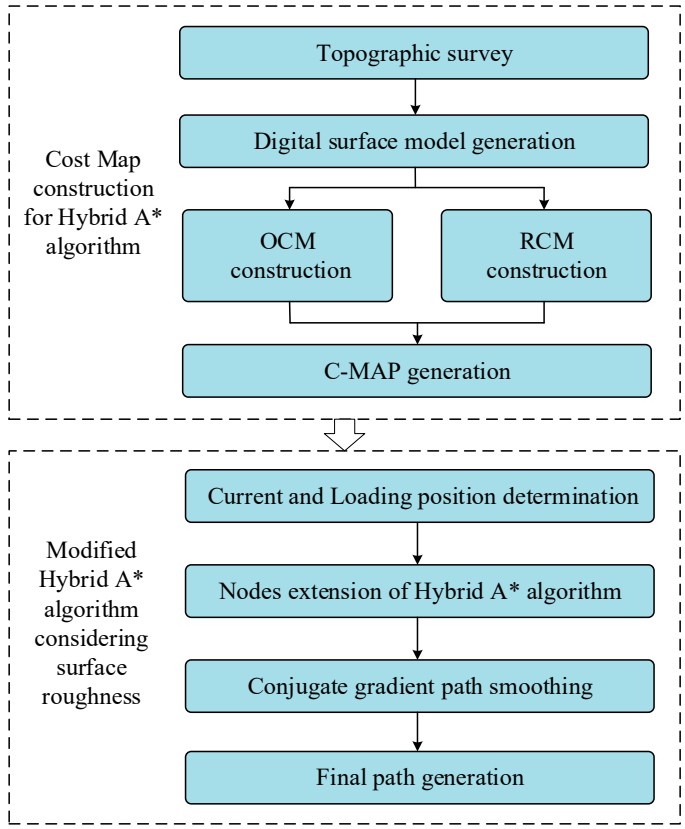

**Figure 1.** The primary process of map construction and Hybrid A* algorithm planning.

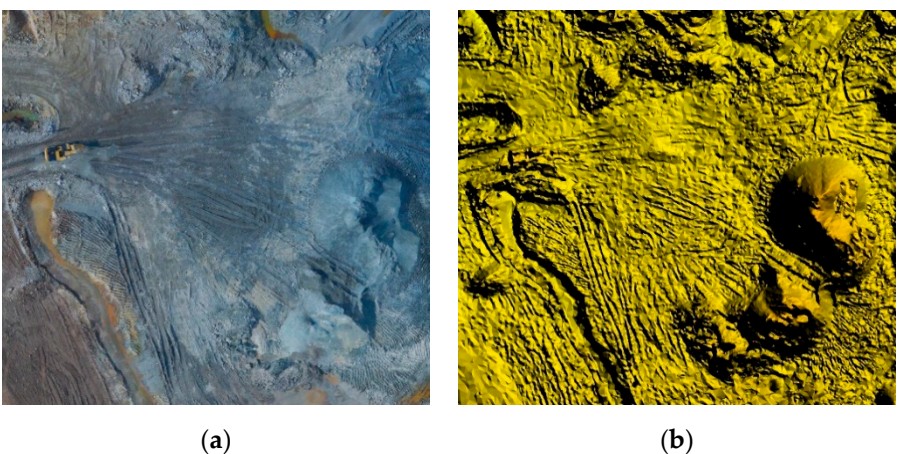

<div align="center">(<b>a</b>)         (<b>b</b>)</div>

**Figure 2.** Positive photograph and corresponding DSM of the cutting zone. (**a**) is the positive photograph; (**b**) is the digital surface model.

### 2.2. Obstacle Cost Map Construction

#### 2.2.1. Elevation Feature Points Extraction

In DSM, the pits or fallen rocks on the ground are expressed in the form of elevation difference. Wang et al. [35] proposed a method to extract the elevation feature points to represent the terrain changes. The feature points were extracted by the four directions (vertical, horizontal, 45° left, 45° right) scanning method based on the section scanning. The key is the selection of terrain thresholds to distinguish different terrains, such as obstacles or little pit. Moreover, for high-precision DSM, Wang pointed out that the equal-interval sampling method can improve the efficiency of subsequent calculations, and the model error is acceptable. Therefore, the original DSM with 0.032 m accuracy is

reprocessed to 0.096 m accuracy by the equal-intervals sampling method in this paper. Referring [16], we set $T = 0.3$ m as the terrain threshold of impassable elevation difference, and $\theta = 15°$ as the slope threshold of the maximum passable slope. The steps of extracting elevation feature points are as follows (take the vertical scanning as an example):

1.  Reforming the elevation matrix $E$ according to the different scanning directions. The elements of each column in $E$ is put into $E_i$, as shown in Equation (2).

$$E_i = \begin{bmatrix} e_{1i} \\ \vdots \\ e_{mi} \end{bmatrix}, E = (E_1, E_2 \dots E_n), i = 1, 2 \dots n \tag{2}$$

2.  Scanning each data point sequentially in $E_i$: set a searching group with $e_{j-1}, e_j, e_{j+1}, (j = 2, 3 \dots m - 1)$, the step length is one. If the point is the first or last point of $E_i$, or local extremum of the group, add the point $p$ to the candidate feature points set $F_p$, where $p$ contains the elevation value and coordinate information. The extracted feature points are shown in Figure 3a.

3.  Filtering the points of $F_p$ by terrain threshold $T$ which is defined by the characteristics of the cutting zone. Then determine whether $p_k$ and $p_{k+1}$ are adjacent in $F_p$. If they are adjacent, judge the absolute value of two-point difference, whether it's over the threshold $T$. If the value greater than $T$, $p_{k+1}$ will be reserved in $F_p$. Otherwise, $p_{k+1}$ will be removed from $F_p$. The original section scanning line and the final feature point set $F_p$ are shown in Figure 3b.

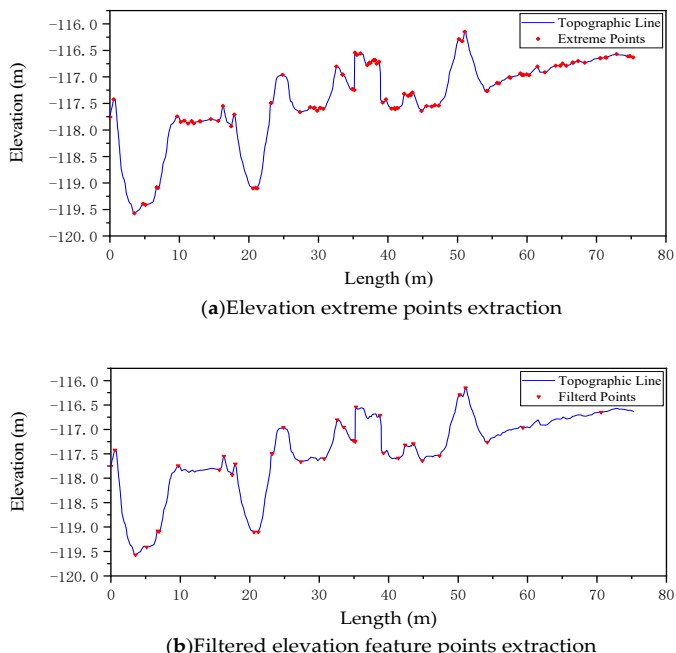

**Figure 3.** Schematic diagram of feature point extraction. In (**a**), the red dots are the extreme points on the section scanning line. In (**b**), the red marks are the feature points filtered according to the threshold $T$, and only the points with large elevation change are retained. The x-axis, which depends on the size of the point set represents the length of the scanline; the y-axis represents the elevation value of the point on the scanline.

### 2.2.2. Binary Map Construction

Binary map construction is the preprocessing work for Obstacle Cost Map construction. Firstly, Compare whether the slope angle $\beta$ of adjacent points $p_k(x_{p_k}, y_{p_k})$ and $p_{k+1}(x_{p_{k+1}}, y_{p_{k+1}})$ in $F_p$ greater

than $\theta$ sequentially. If so, all the corresponding coordinate points between $p_k$ and $p_{k+1}$ are assigned as 1, otherwise, it is 0, i.e., the value of one is represented obstacle. As shown in Equations (3) and (4). Figure 4 is the schematic diagram of the four-direction scanning binary map.

$$\beta = tan^{-1}\left(\frac{y_{p_{k+1}} - y_{p_k}}{x_{p_k} - x_{p_k}}\right), \tag{3}$$

$$p_j = \begin{cases} 1, \beta \geq \theta \\ 0, \beta < \theta \end{cases}, j = 1, 2 \ldots m, \tag{4}$$

where $p_j \in E_i$.

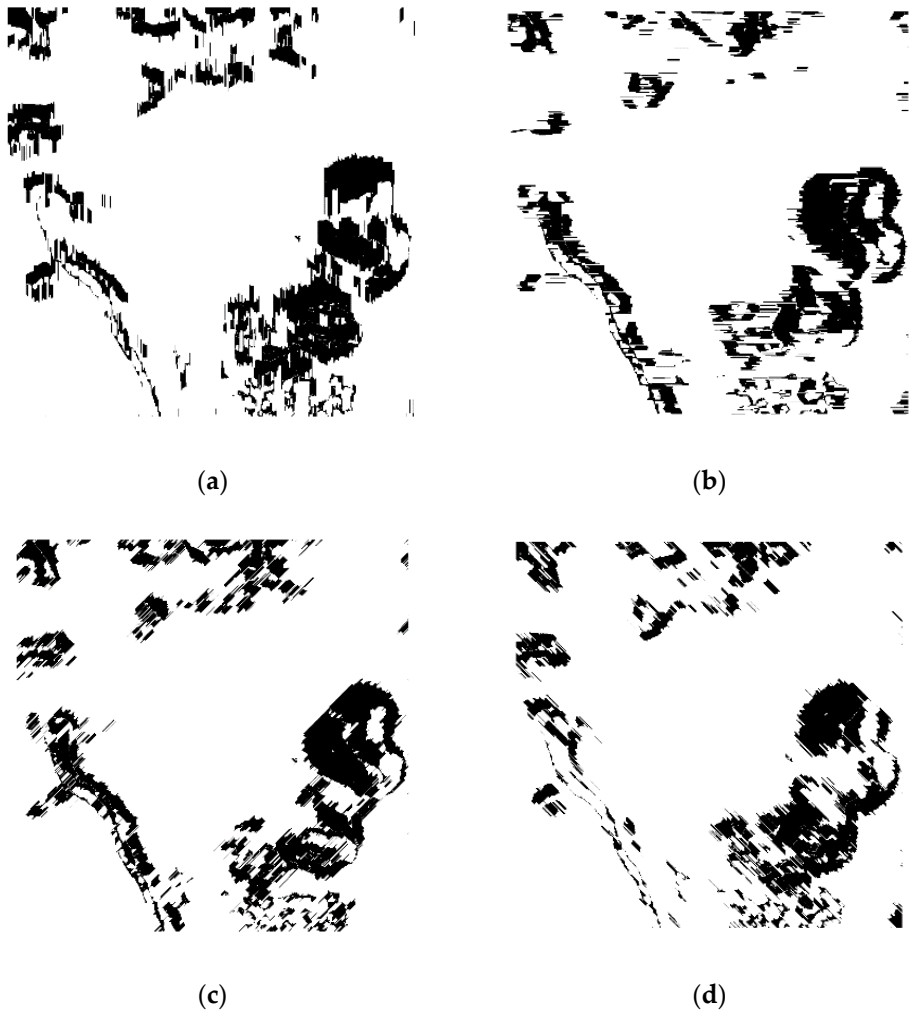

(a)

(b)

(c)

(d)

**Figure 4.** Schematic diagram of four direction scanning binary map. (**a**) is the vertical scanning map; (**b**) is the horizontal scanning map; (**c**) is the 45° left scanning map; (**d**) is the 45° right scanning map.

Secondly, fuse four scanning binary maps to form the final binary map in a simple mixture method: Traverse all points of the four maps. If the accumulated value of the same coordinates in four maps is greater than or equal to 2, then the value of the coordinate point is set to 1. Otherwise, it is 0. The final obstacle binary map is shown in Figure 5.

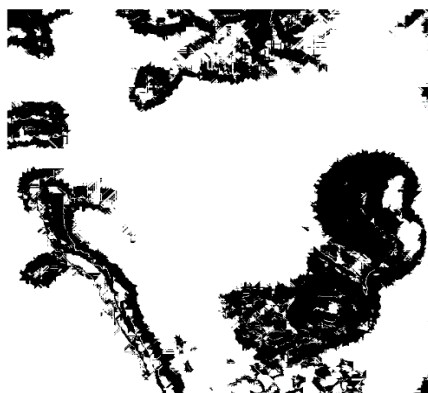

**Figure 5.** Obstacle binary map of cutting zone. The black area is the obstacle area, which retains the typical details of the four-direction scanning binary maps.

### 2.2.3. Obstacle Cost Map Generation

For the heuristic algorithm, the map cost value directly affects the accuracy of the estimation function, thus affecting the quality of the final path. Voronoi diagram is a geometry-based route planning method for robot path planning [36,37], which divides the space into several regional units according to the nearest neighbor attributes of the elements in the site collection. Dolgov [30], the inventor of the Hybrid A* algorithm, used the generalized Voronoi diagram to generate the planning map, which made the planning path can pass through the narrow area and without collision. The generation steps for the obstacle cost map are as follows:

1. Building the two-dimensional Voronoi diagram from the binary map. The 1-value coordinate points in the binary map of obstacles are regarded as the scatter points. Then, calculate the 2-D Voronoi diagram of all scatter points. The red dot is the Voronoi station, and the blue line is the Voronoi edge, as shown in Figure 6a.
2. Cutting off the Voronoi edges which passing through the obstacle polygon, and the remaining edges are the edges of GVD. The black square is the obstacle polygon, and the red line is the GVD edges, as shown in Figure 6b.
3. Converting the GVD into grid form by sampling method, as shown in Figure 6c. Then, calculating each grid cost by Equation (5) to construct the obstacle cost map. The grid cost reaches its maximum within obstacles, and the color is close to black. It reaches its minimum on the GVD edges. The color is close to white, as shown in Figure 6d.

$$gridCost(x,y) = \begin{cases} \left(\frac{\alpha}{\alpha+d_o(x,y)}\right)\left(\frac{d_v(x,y)}{d_o(x,y)+d_v(x,y)}\right)\left(\frac{(d_o-d_o^{max})^2}{(d_o^{max})^2}\right), & d_o < d_o^{max} \\ 0, & others \end{cases} \tag{5}$$

where $gridCost(x,y)$ is the cost of each grid. It is continuously distributed in the range of 0 to 1, only takes 1 in the obstacles, and takes 0 on the Voronoi edge. $\alpha$ is the parameter to control the descent rate of cost, $\alpha > 0$. $d_o^{max}$ is the maximum range of potential value distribution, $d_o^{max} > 0$. For the grid points that exceed the $d_o^{max}$, the grid cost is 0. $d_o$ is the distance from the grid point to the nearest obstacle edge. $d_v$ is the distance from the grid point to the nearest Voronoi edge.

Using this method to process the obstacle binary map of the cutting zone, we get the corresponding obstacle cost map, as shown in Figure 7. However, due to the high grid precision and the maximum range of potential value distribution, only the area at the edge of the obstacle (black area) has an available cost (gray area). For the rest of the open space area (far away from the obstacle), the grid cost is 0 (white area). Therefore, to make the algorithm consider the cost in the open area, we construct the roughness cost map to complement this area.

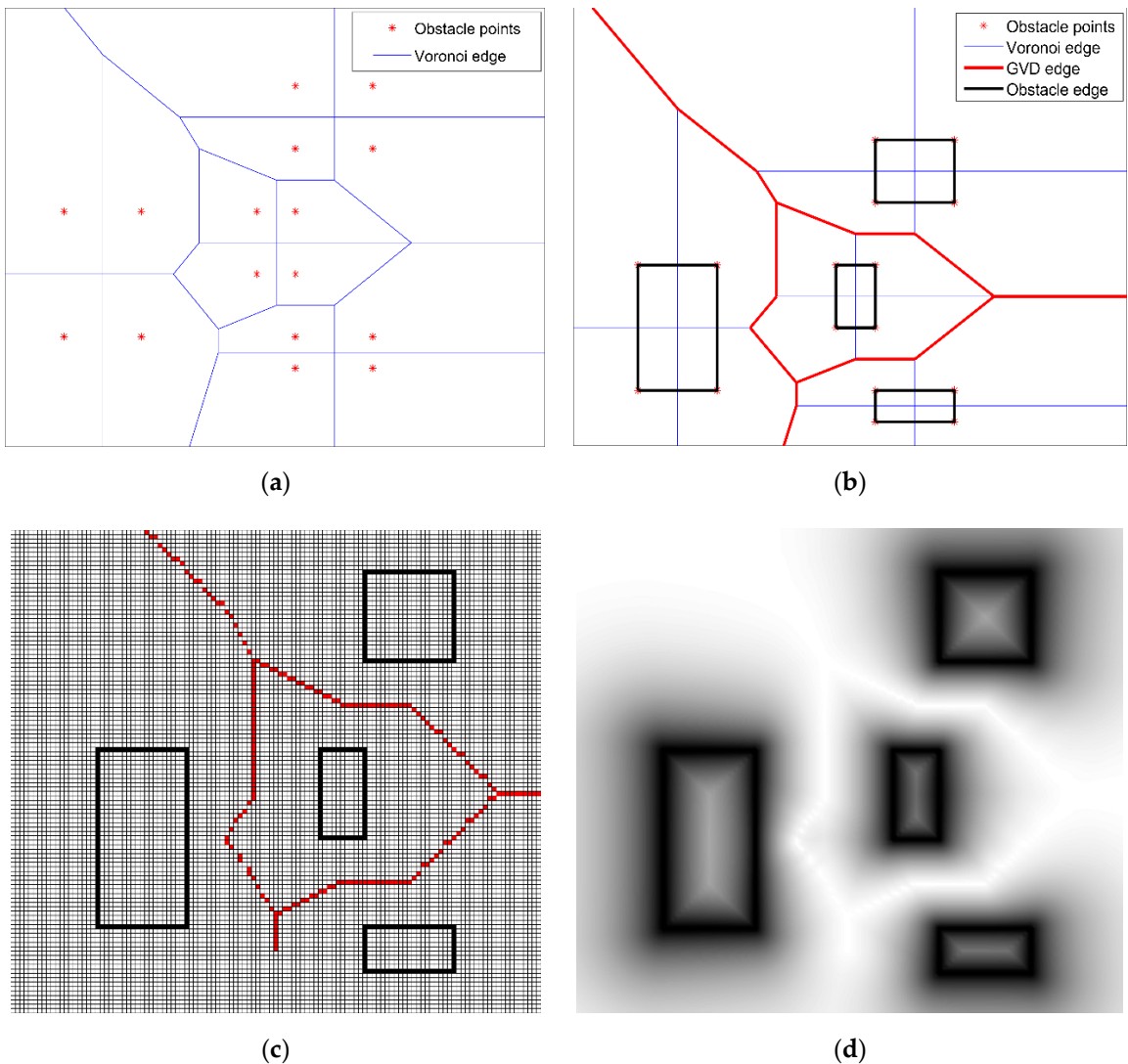

**Figure 6.** Schematic diagram of Obstacle Cost Map generation. (**a**) is the original Voronoi diagram; (**b**) is the GVD; (**c**) is the grid form of GVD; (**d**) is the Obstacle Cost Map, the black color represents the obstacle grid and the white represents the Voronoi edge.

### 2.3. Roughness Cost Map Construction

A large number of blank areas exist in the obstacle cost map of the cutting zone, in which there are many possible adverse road conditions. If it is not included in the heuristic function of the planning algorithm, it will result in a deviation of the planning path. Generally, the standard deviation of the sliding window can be used to estimate the ground surface roughness [38]. The smaller the standard deviation value, the smoother the road surface. We use this method to construct the roughness cost map:

1.  Removing the coordinate of obstacles in the original elevation point set $E$, and obtained a new point set $E_r$, which only keeps the elevation value of the open space area.
2.  Using the section scanning method (threshold $T = 0$, $\theta = 5^\circ$) to process the open space area, deleting the elevation value of a point below the threshold from $E_r$.
3.  In $E_r$, taking a window and using the sliding standard deviation method to traverse $E_r$, and calculating the standard deviation of the window, then assigning its value back to each point in this window.

4.  Normalizing all the values of points in $E_r$ with a range of 0 to 1, which is the same as the obstacle cost range.

5.  Through the above steps, we can get the rough ground cost map of the cutting zone, as shown in Figure 8a.

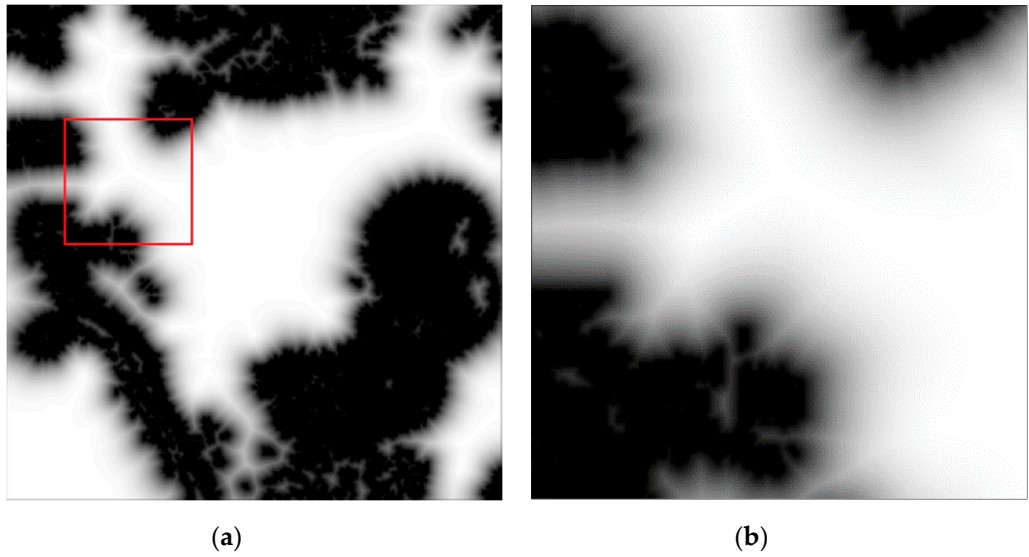

(**a**)                                         (**b**)

**Figure 7.** Schematic diagram of the open space. (**a**) is the obstacle cost map of cutting zone; (**b**) is the open space area in the red square in (**a**).

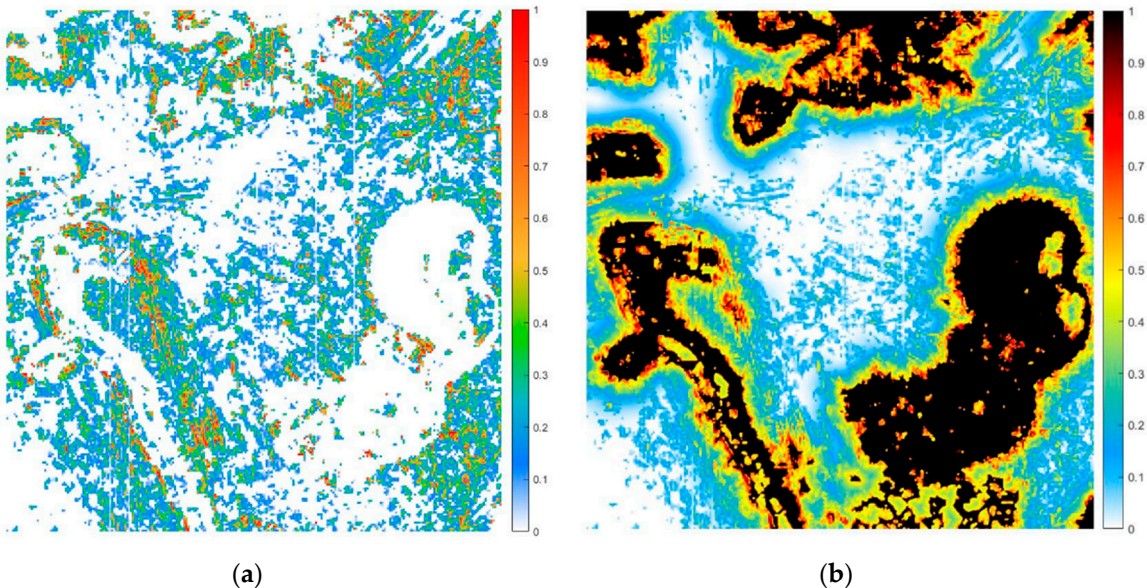

(**a**)                                         (**b**)

**Figure 8.** Schematic diagram of Roughness Cost Map and C-MAP. (**a**) is the Roughness Cost Map without obstacle; (**b**) is the C-MAP. The color distribution represents the traversable of the road surface. Black represents obstacle area.

### 2.4. C-MAP Generating

After constructing OCM and RCM with the same value range at $cost = [0, 1]$, the C-MAP can be constructed as follows:

1.  Removing the obstacle points (cost = 1) from OCM, and adding the remaining OCM to RCM.

2.  Normalizing the obtained matrix.

3.    Backfilling obstacle points of OCM.

The constructed C-MAP still maintain the value range at $cost = [0, 1]$, and it contains both the obstacle information (black part of Figure 8b) and surface roughness information in the open space area (other color parts of Figure 8b), in which the planning algorithm can take roughness cost into account in the estimation function.

*2.5. Hybrid A\* Algorithm Improvement*

This section introduces the planning process of the modified Hybrid A\* algorithm, and how to optimize it to fit the scene described in this article. The Hybrid A\* algorithm is an improved A\* path planning heuristic algorithm proposed by Dolkov [30], which adds the vehicle non-holonomic constraints to the estimation function. Based on the two-dimensional grid $(x, y, \theta)$ extension of A\* algorithm, it adds a three-dimensional search, considers the three-dimensional state $(x, y, \theta, \lambda)$ of the vehicle. Where $x$ and $y$ are coordinate information, $\theta$ is direction information, and $\lambda$ is vehicle forward and backward status information. Nevertheless, the path produced by the Hybrid A\* algorithm has local bending, so Dolkov uses the conjugate gradient (CG) method to smooth the path.

We add the cost of tire contact with the road surface into the estimation function of the Hybrid A\* algorithm instead of only taking the cost of the center point as the estimation. Also, we replace the Voronoi term with the Roughness term to enhance the path's avoiding ability for roughness surface.

2.5.1. Node Extension Improvement

The node extension of Hybrid A\* is guided by two heuristic methods. One is the 2D heuristic method in the traditional A\* algorithm, which considers the information of obstacles in the map, but ignores the physical attributes of the vehicle itself. Its node extension only contains coordinate information and direction information, as shown in Figure 9a. Another is the non-holonomic-without-obstacles heuristic method. This method considers the vehicle non-holonomic constraints and assumes that there are no obstacles in the map, assumes the goal state of $(x_g, y_g, \theta_g) = (0, 0, 0)$, computes the shortest path to the goal from every point $(x, y, \theta)$. The schematic extension diagram is shown in Figure 9b. The second method uses the maximum of the two methods cost as the heuristic cost. Also, in order to save computing time, the algorithm uses Reed-Shepp (RS) curve to judge the current node and the goal node periodically. If there is no obstacle between the two points, an obstacle-free RS path will be generated directly between the two points. We set a few parameters, such as number of motion primitives (NMP), motion length (ML), and extension interval (EI), to control the node extension. NMP determines how many next nodes will be calculated based on the current node, by way of example, in Figure 9b, the NMP is 5, so there are five forward and five backward points. ML is the length of each extension determined by the minimum turning radius. EI determines how many loops to perform RS curve detector.

When the Hybrid A\* algorithm estimates the moving cost, it usually takes the grid value of the center point as the estimation cost. We improve the $g$-value estimation function of the Hybrid A\* algorithm, add a new term to estimate the cost of tire contact with the ground when the vehicle is moving. The improved node extension diagram is shown in Figure 10, the yellow area is the coverage area of the tire when extending, and the occupancy grid value within the range is taken as the extension cost.

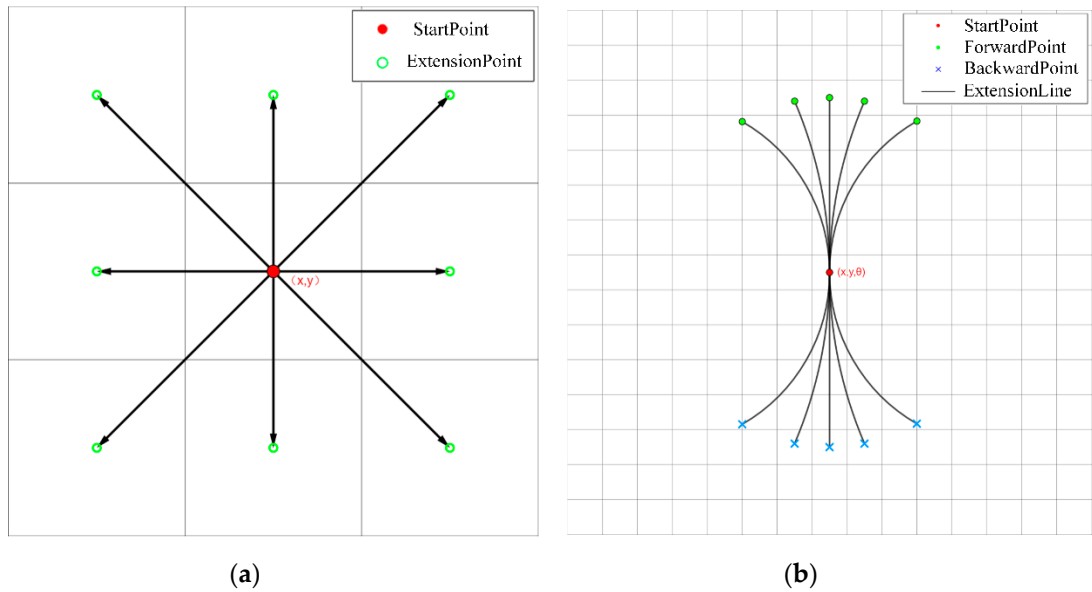

**Figure 9.** Schematic diagram of node extension. (**a**) is the traditional A* algorithm node extension; (**b**) is the nonholonomic constraint node extension of Hybrid A* algorithm. The central red dot is the current node and the green or blue node is the calculated next node.

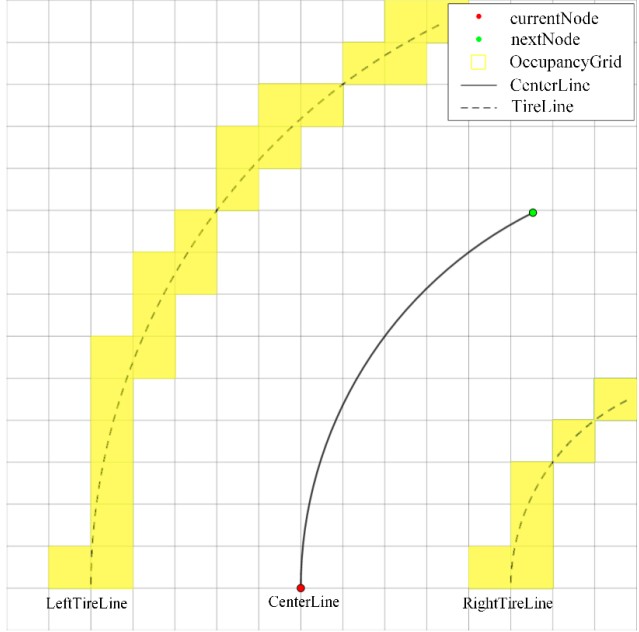

**Figure 10.** Schematic diagram of the tire-contact cost estimation. The black solid line is the extension line from the current node to the next node. The dashed is the tire path calculated from vehicle properties. The yellow area is the occupied grid of the tire path.

### 2.5.2. Estimation Function Improvement

The Hybrid A* algorithm uses the Equation (6) to estimate the cost of each node extension.

$$f(x) = g(x) + h(x), \tag{6}$$

where the first term $g(x)$ is the sum of the estimated cost from the start to the current node, the second term $h(x)$ is the heuristic value from the current node to the goal, and the sum of the two terms is the evaluation function of the whole algorithm. This paper only improves the $g(x)$, and use the two

heuristic functions of the original algorithm to calculate the $h(x)$. The calculation of $g(x)$ is realized by Equation (7).

$$g(x) = DirectionCost \times (g(x_{parent}) + length(x_{parent}, x) + TireCost(x_{parent}, x)) + \lambda \cdot SwitchCost \quad (7)$$

where *DirectionCost* contains ForwardCost (FC) and ReverseCost (RC), penalizing the motion cost according to a different direction. $g(x_{parent})$ represents the sum cost from the start point to the parent node of the current node. $length(x_{parent}, x)$ is the length of the curved path between the parent node and the current node. *SwitchCost* is the direction switch cost to penalize the motion state, when the direction of the vehicle is changed $\lambda = 1$, otherwise $\lambda = 0$. We add a new cost term, $TireCost(x_{parent}, x)$, based on the original $g$-value estimation function, to estimate the cost of tire contact with the surface. This term represents the sum cost of the occupied grid values by the tire line from the current node to its parent node:

$$TireCost(x_{parent}, x) = \sum_{i=x_{parent}}^{x} (CostMap_i) + \sum_{j=x_{parent}}^{x} (CostMap_j), \quad (8)$$

where the first term is the accumulative cost of the left tire, and the second is the right.

### 2.5.3. Conjugate Gradient Method Improvement

There are still some unnecessary turns in the path planned by the Hybrid A* algorithm. Dolkov used the conjugate gradient (CG) smooth method to optimize the path. The objective function is:

$$P = \omega_\rho \sum_{i=1}^{N} \rho v(x_i, y_i) + \omega_o \sum_{i=1}^{N} \sigma_o(|x_i - o_i| - d_{max}) + \omega_\kappa \sum_{i=1}^{N-1} \sigma_\kappa(\frac{\Delta\Phi_i}{|\Delta x_i|} - \kappa_{max}) + \omega_s \sum_{i=1}^{N-1} (\Delta x_{i+1} - \Delta x_i)^2, \quad (9)$$

where the first term is Voronoi term to guide the path away from an obstacle in both narrow and wide passages, the second term is Obstacle term to penalize the collisions, the third term is curvature term to ensure the path curvature satisfies vehicle non-holonomic constraints, and the fourth term is the smoothness term to make the path smoother.

In the cutting zone scene, there is a large open space area instead of narrow passages. Thus, we replace the Voronoi term with roughness term in CG, and the function is:

$$P_r = \omega_r \sum_{i=1}^{N-1} \sigma_r \left( \sum_{n=1}^{4} \omega_n(|x_{in} - r_{in}|) \right) \quad (10)$$

The derivative is:

$$\frac{\partial P_r}{\partial x_i} = 2\omega_r \cdot \left( \sum_{n=1}^{4} \omega_n(|x_{in} - r_{in}|) \right) \cdot \left( \sum_{n=1}^{4} \omega_n \frac{x_{in} - r_{in}}{|x_{in} - r_{in}|} \right), \quad (11)$$

$$\omega_n = 1 - \left( r_{nmax} / \sum_{n=1}^{4} r_{nmax} \right), \quad (12)$$

where $\omega_r$ is the weight of gradient; $\sigma_r$ is a simple quadratic penalties function; $x_{in}$ are the locations of four tire contact with the surface which corresponding to each path point; $r_{in}$ are the locations of the maximum value roughness points nearest to the $x_{in}$; $r_{nmax}$ is the distance between $r_{in}$ and $x_{in}$; $\omega_n$ is the weight of the gradient.

This term penalizes the contact between the tire and the nearby rough ground, which increases the ability to deviate from rough terrain during path point optimization.

2.5.4. The Modified Hybrid A* Algorithm in Detail

The modified Hybrid A* algorithm considering terrain roughness is improved on the regular Hybrid A* algorithm, which has the same essential parts. Before planning starts, it is necessary to set vehicle parameters, including length, width, tire information, and minimum turning radius, etc. Generally, tire information includes information on wheelbase and tire size. After that, the C-MAP is input, which includes both obstacle and roughness information to the path planner, and then the essential planning parameters are set, e.g., FC, RC, EI, etc. The pseudo code for Modified Hybrid A* algorithm (Algorithm 1) is as follows.

---

**Algorithm 1** Modified Hybrid A*

---

1:    Set Vehicle properties:
      $VP \leftarrow (Length, Width, Tire\ information, Minimum\ turning\ radius)$
2:    Set planner: $planner \leftarrow (CostMap, Planning\ parameters(ML, NMP, FC, RC, SC, EI))$
3:    Input: $startNode(x_s, y_s, \phi_s), goalNode(x_g, y_g, \phi_g)$
4:    **Procedure** Hybrid A* planner $(startNode, goalNode, VP, planner)$
5:    Node extension
6:    Validate the node
7:    RS curve check
8:    **return** Path Points
9:    Processing Path Points by ConjugateGradientSmoother $(CostMap, PathPoints)$
10:   **return** the Path Points after CG optimization
11:   **end procedure**
12:   **Procedure** gScore $(n_{current}, n_{next}, planner, VP)$
13:   $gScore_{parent} \leftarrow gScore$ of $n_{current}$
14:   $Length_{current \rightarrow next} \leftarrow ML$
15:   Generate the tire path besides the extension line according to $VP$
16:   Calculate the occupied grid coordinates of the tire path
17:   Delete duplicate grid coordinates
18:   $TireCost$ is the total cost of the occupied gird
19:   **if** the direction of $n_{next} \neq$ the direction of $n_{current}$ **then**
20:      $gScore = gScore_{parent} + Length_{current \rightarrow next} + TireCost + SC$
21:   **else**
22:      $gScore = gScore_{parent} + Length_{current \rightarrow next} + TireCost$
23:   **end if**
24:   **end procedure**

---

The procedure gScore is an improved *g*-value estimation function. In each node extension step, it accumulates the total cost of the occupied grid of the corresponding tire path which is calculated from vehicle parameters. This allows Hybrid A* algorithm can incorporate surface roughness into motion cost estimation. Steps 19 to 23 mean that only when the motion status changes, in other words, when the motion direction changes from forward to backward, the SwitchCost will penalize the motion cost.

## 3. Performance Analysis of Modified Algorithm

To verify the effectiveness and robustness of the modified algorithm, we experiment with and analyze its performance in two different scenes of the Chengmenshan copper mine in Jiangxi. The original image data of the cutting zone are obtained by quad-rotor UAV oblique photogrammetry (Figure 11), and the corresponding C-MAPs of two scenes are constructed by the method proposed in this paper (Figure 12). In these two scenes, we set three different start positions and a fixed position of truck for path planning and verification. Path length, accumulative path cost, and path total score are used for the path evaluation.

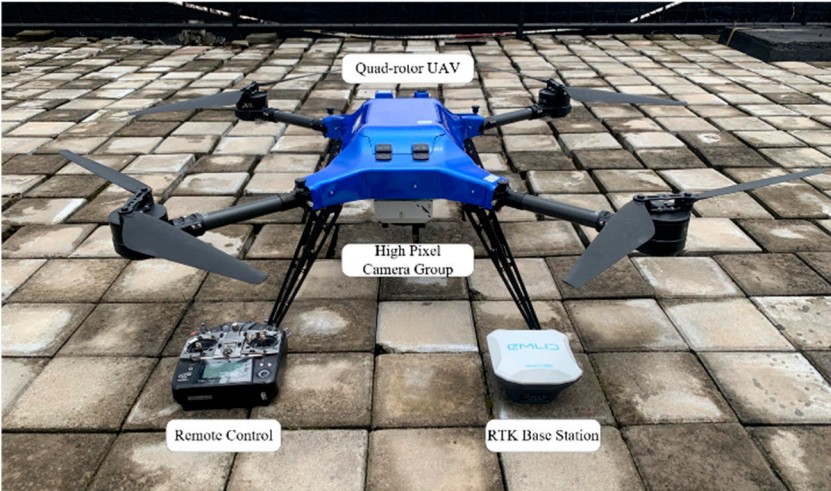

**Figure 11.** Quad-rotor UAV.

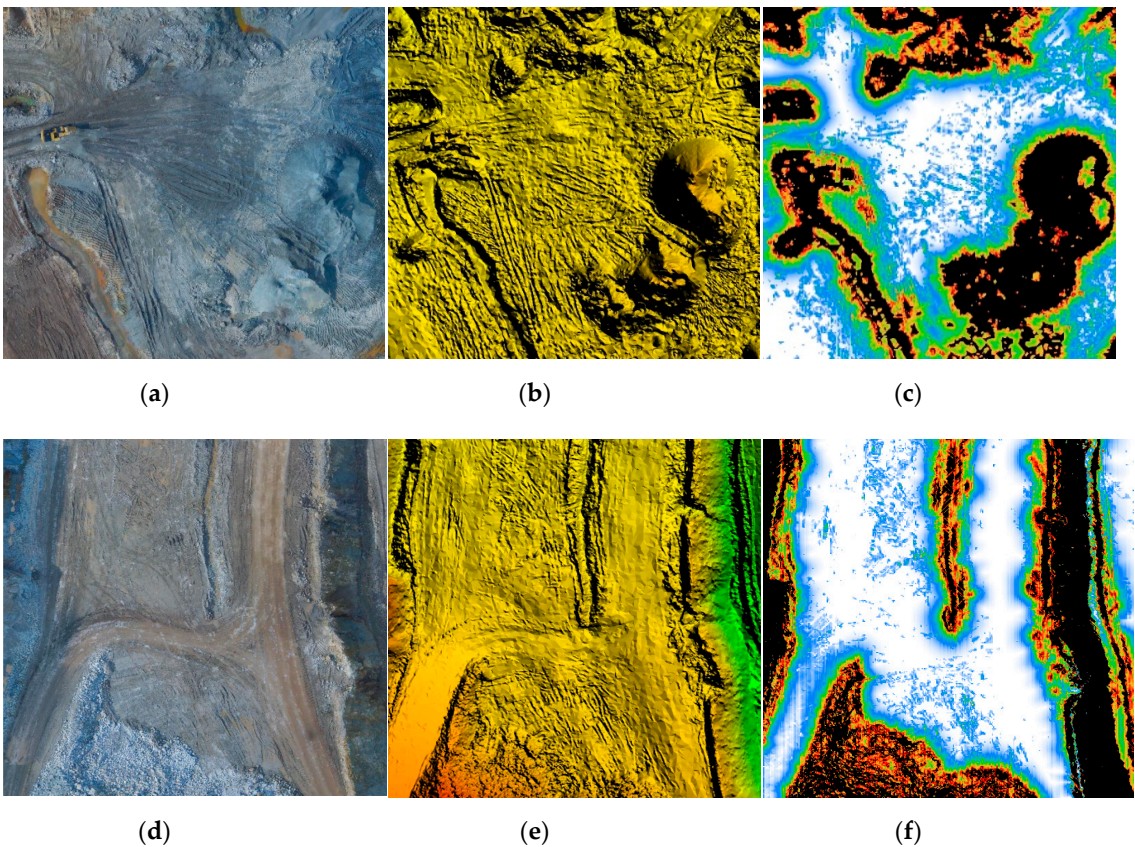

**Figure 12.** Schematic diagram of two cutting zone scenes. (**a**,**d**) are the orthophotographs; (**b**,**e**) are the DSM; (**c**,**f**) are the C-MAP.

Table 1 is the cartographic information, and Table 2 is the parameters of the simulated haulage truck. Table 3 shows the main parameters of the Hybrid A* algorithm. The parameters of conjugate gradient smoother are set to $\omega_r = \omega_o = 0.002$, $\omega_\kappa = \omega_s = 0.2$.

**Table 1.** Cartographic information.

| Camera Pixels of UAV | DSM Original Accuracy | DSM Accuracy after Sampling | C-MAP Unit Size |
|---|---|---|---|
| 20 megapixel × 5 | 0.032 m | 0.096 m | 0.1 m × 0.1 m |

**Table 2.** Simulated haulage truck parameters.

| Length | Width | Wheelbase | Minimum Turning Radius | Tire Size |
| --- | --- | --- | --- | --- |
| 8.7 m | 4.525 m | 3.75 m | 7.2 m | 18.00-33-32 PR |

**Table 3.** Hybrid A* algorithm parameters.

| Number of Motion Primitives | Motion Length | Forward Cost | Reverse Cost | Switching Cost | Extension Interval |
| --- | --- | --- | --- | --- | --- |
| 5 | 72 | 1 | 5 | 100 | 30 |

### 3.1. Analysis of Algorithm Capability in Different Scenes

According to the actual mining haulage route, we demarcate three initial starting positions in the green square frame, and a loading position in the red square frame, as shown in Figure 13. Then we verified the algorithm by the following two parts:

1. Testing the recognition and avoidance ability of modified Hybrid A* algorithm for rough terrain. Firstly, plan a path in the obstacle cost map without roughness information. Then, after a virtual rough area is artificially added to the map and plan the path again. The result is shown in Figure 14.

2. Evaluating the path planned by the modified Hybrid A* algorithm in obstacle cost map and C-MAP, the results are shown in Figure 15.

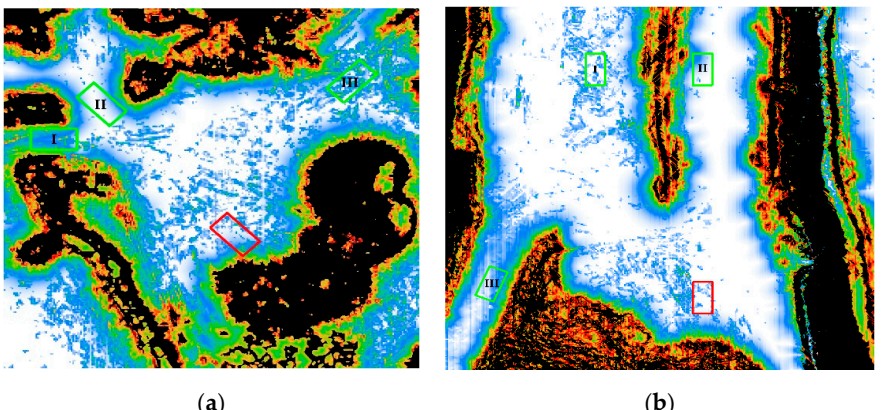

(**a**)        (**b**)

**Figure 13.** Schematic diagram of start and goal position of truck in two scenes. (**a**) is the map of scene 1; (**b**) is the map of scene 2. The green square frame represents the starting position and the red square is the goal position.

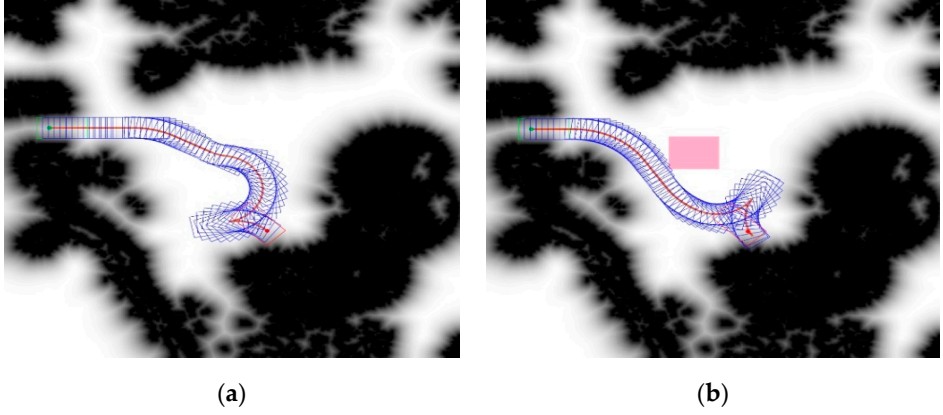

(**a**)        (**b**)

**Figure 14.** Schematic diagram of the roughness avoidance ability of the algorithm. (**a**) is the planning path in the OCM; (**b**) is the planning path in the OCM with the virtual rough area (pink block).

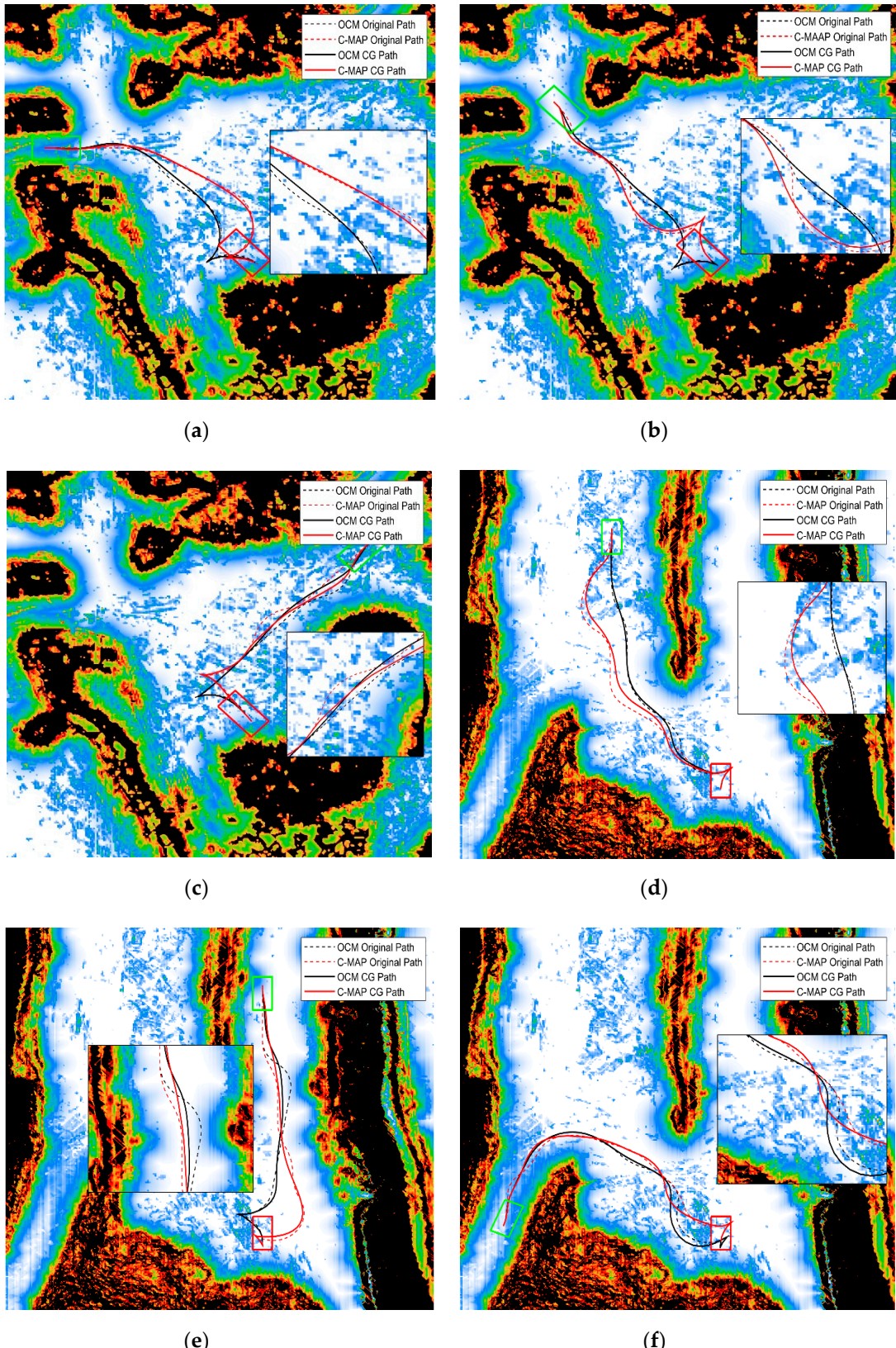

**Figure 15.** Verification of path planning in two scenes. (**a**–**c**) are the path planning experiments in scene 1; (**d**–**f**) are the path planning experiments in scene 2.The black line represents the path planned in OCM, and the red line represents the path planned in C-MAP. The dotted line is the path without CG optimization, and the solid line represents the path after CG optimization.

In the first part of the test, it is evident that after adding the virtual rough area in the obstacle cost map, the path avoids the rough area. It is proved that the modified Hybrid A* algorithm can detect and avoid the large roughness.

In the second part of the test, it is intuitive to see in the comparison diagram that the red line deviates from the rough area better than the black line. In order to quantitatively evaluate the different paths in two scenes, we measure the path length and the accumulative path cost of the tires on both sides.

Since it is challenging to integrate the path length and accumulative cost into actual vehicle operating cost, we use the entropy method [39] to calculate the weight and get the total score of each path to approximate the operating cost with the Equations (13) and (14).

$$E_j = -ln(n)^{-1} \sum_{i}^{n} p_{ij} ln(p_{ij}),$$

(13)

$$\omega_i = \frac{1 - E_j}{k - \sum E_j} (i = 1, 2, \ldots, k),$$

(14)

where $p_{ij} = Y_{ij} / \sum_{i=1}^{n} Y_{ij}$, $Y_{ij} = \frac{X_{ij} - min(X_i)}{max(X_i) - min(X_i)}$, $X_i = \{length, cost\}$. $X_i$ is a matrix of path length and accumulative cost information for each path.

In Table 4, the optimization rate refers to the optimization degree of the total score of C-MAP compared with the total score of OCM. The results of path evaluation Table 4 show that the planning path with consideration of roughness is better than that without consideration. In these three scenes, the optimization rate is more significant than zero, which shows that the modified algorithm can reduce the potential cost of the planning path.

**Table 4.** Planning Path Evaluation Table in Two Scene.

|  |  | Path Length (m) | Accumulative Cost | Total Score | Optimization Rate (%) |
|---|---|---|---|---|---|
| Scene 1 | OCM | 40 | 1124.4 | 1113.8 | 5.93 |
| Start I | C-MAP | 43 | 1057.7 | 1047.8 | |
| Scene 1 | OCM | 39 | 926.9 | 918.2 | 5.26 |
| Start II | C-MAP | 37 | 878.1 | 869.9 | |
| Scene 1 | OCM | 41 | 316.4 | 313.7 | 6.87 |
| Start III | C-MAP | 39 | 294.6 | 292.1 | |
| Scene 2 | OCM | 57 | 386.2 | 382.9 | 18.23 |
| Start I | C-MAP | 61 | 315.7 | 313.1 | |
| Scene 2 | OCM | 54 | 1174.8 | 1163.6 | 9.82 |
| Start II | C-MAP | 58 | 1059.4 | 1049.3 | |
| Scene 2 | OCM | 62 | 1096.8 | 1086.4 | 6.99 |
| Start III | C-MAP | 59 | 1020.1 | 1010.5 | |

## 3.2. Analysis of Algorithm Robustness

To test the optimization of the modified Hybrid A* algorithm under random conditions, we have conducted thousand planning experiments in two scenes with randomly given start position and goal position. We are moreover repeat the experiment at three different terrain resolutions scenes (0.1 m, 0.5 m, 1.0 m). Figure 16 is the schematic diagram of two scenes with lower resolution.

With the decrease of terrain resolution, C-MAP loses a lot of surface roughness information. As a result, in the planning of low-resolution maps, the optimization rate is more concentrated in the 0% to 10% range, as shown in Figure 17c–f. Significantly, when the resolution is from 0.1 m to 0.5 m, the average optimization rate of two scenes decreases rapidly, while the latter decreases slowly from 0.5 m to 1.0 m, as shown in Figure 18. The result shows that the planning algorithm cannot optimize the trajectory well when the terrain resolution (grid accuracy) is reduced from high to low.

Based on the results of the random test, we consider several factors that affect the optimization rate:

1.  The influence of planning map construction, including the DSM accuracy and filter threshold. The accuracy of DSM directly affects the recognition ability of the construction algorithm for surface roughness. However, the improvement of accuracy will result in more computational work and the program needs to process more data units. Therefore, 0.1 m is an appropriate accuracy value.

2.  For the filter threshold, it affects the ability of the construction algorithm to distinguish obstacles from the surface. Moreover, it is difficult to construct a surface roughness model if the terrain is flat.

3.  The influence is caused by the parameter setting of Hybrid A* algorithm. The setting of parameters will directly affect the quality of the final path. To illustrate, if the ReverseCost setting is greater than ForwardCost, the algorithm will consider the backward situation more. If the Extension Interval setting is too large, it will cause extra computation. Switch Cost penalizes the direction change of a vehicle. However, in this paper, since Tire Cost is a dynamic value, the static value of Switch Cost will weaken its effect so that it can be considered as an adaptive value.

4.  Caused by statistics of path cost. The entropy method may not be able to integrate the path length and the accumulative cost on the surface very well, which makes the calculation error.

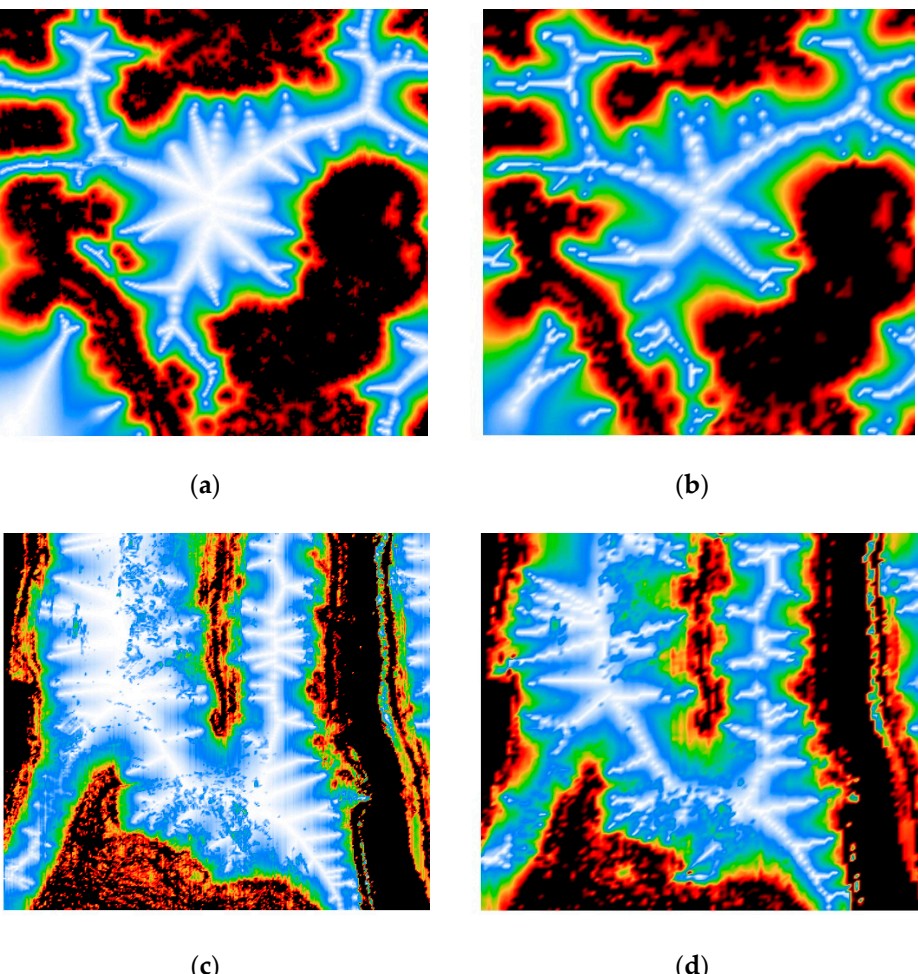

(a)          (b)

(c)          (d)

**Figure 16.** Two scenes with different terrain resolution. (**a,c**) are the 0.5 m terrain resolution scenes; (**b,d**) are the 1.0 m terrain resolution scenes.

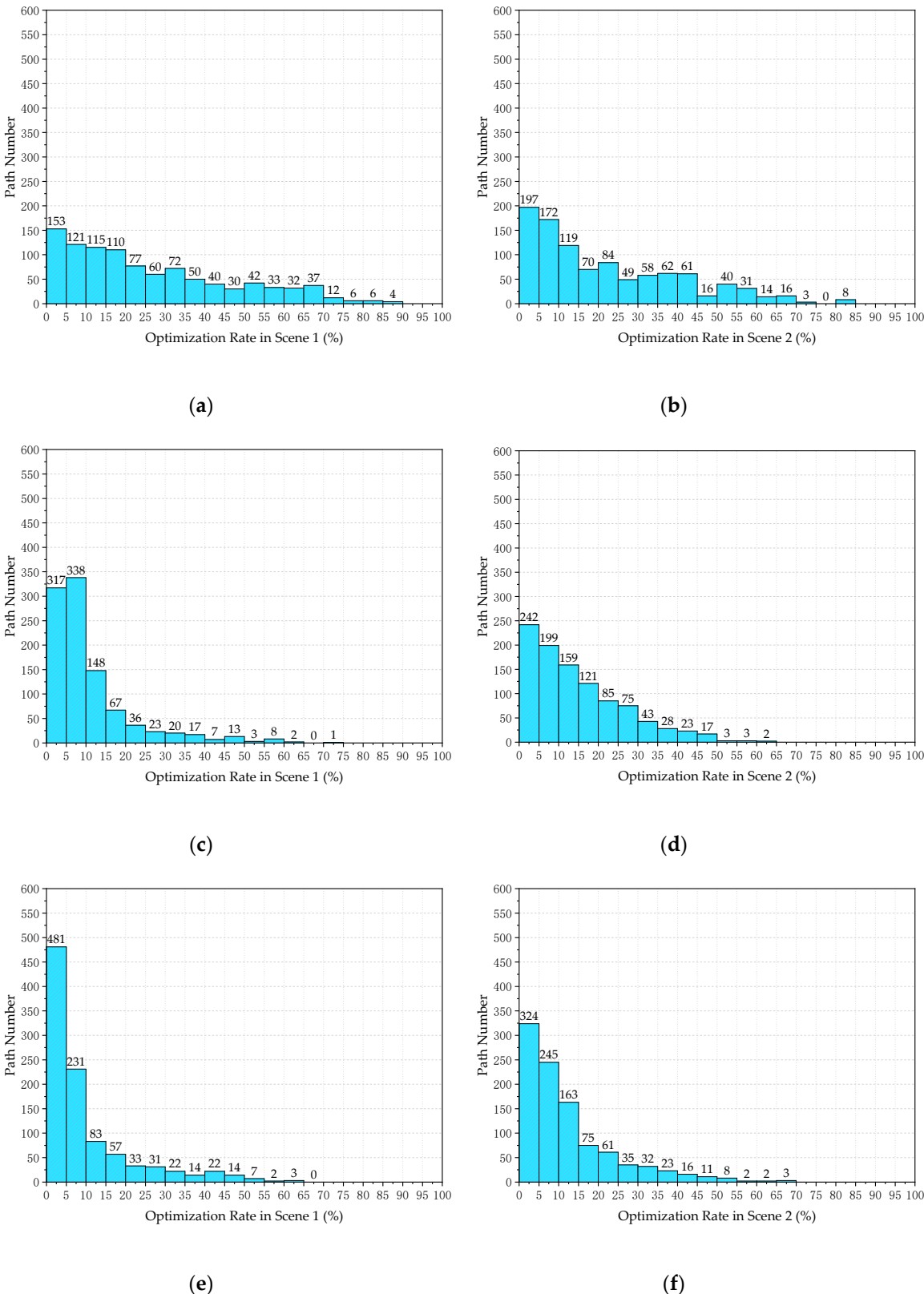

**Figure 17.** Optimization rate of the random path in two scenes with different terrain resolutions. (**a**,**c**,**e**) are the random tests in scene 1; (**b**,**d**,**f**) are the tests in scene 2. (**a**,**b**) are tested in the 0.1 m terrain resolution scenes; (**c**,**d**) are tested in the 0.5 m terrain resolution scenes; (**e**,**f**) are tested in the 1.0 m terrain resolution scenes.

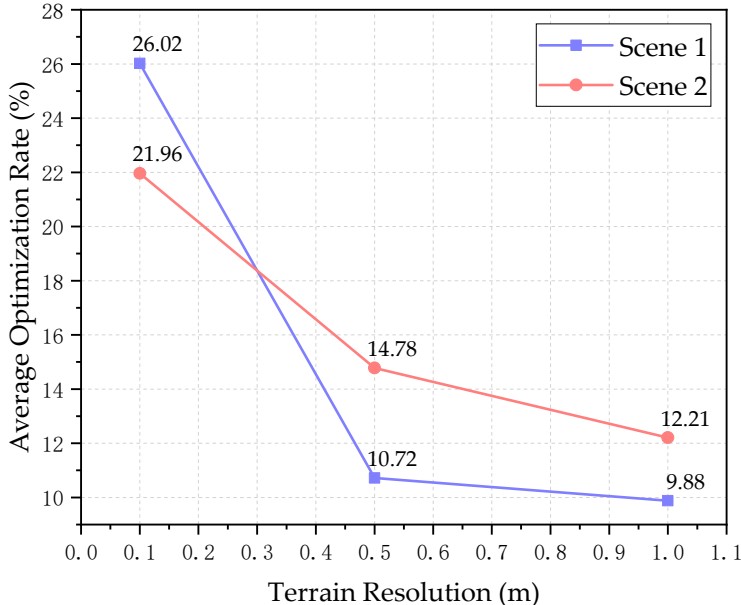

**Figure 18.** Average optimization rate with different terrain resolution. The label under each point is the average optimization rate at that resolution.

## 4. Conclusions

The trajectory planning of cutting zone for unmanned trucks is essential in the autonomous transportation of open-pit mines. When planning the path, if we only consider the obstacle information, the roughness of ground may lead to extra tire wear. This paper proposes a path planning method considering terrain factors. (1) generate the DSM of the cutting zone using the UAV oblique photogrammetry method. (2) based on the DSM, construct a high-precision cost map (C-MAP) which contains both obstacle and roughness terrain information. (3) On the basis of C-MAP, plan a path by the modified Hybrid A* algorithm that incorporates the surface roughness information into the $g$-value estimation function. The random tests of actual scenes demonstrate a 10% to 20% reduction of the path cost. Therefore, our proposed method can reduce the transportation cost of the haulage truck and solve the problem of high cost in the last section of open-pit transportation.

## 5. Future Work

Due to the variability of the cutting zone environment, the method is only applicable to the global static map before transportation. During the mining operation, local elevation will change. Therefore, future work can be divided into two parts:

1. We will collect the environmental information of the cutting zone and construct a real-time local DSM map by using detection sensors, such as Lidar, camera, GNSS receiver, etc. This enables continuous path planning by continually updating the global map and maintaining its timeliness and accuracy. Due to the errors in acquisition equipment and mapping procedure, the matching and fusion updating of the real-time local map and the global map is the focus and difficulty of future works.
2. We will study the relationship between road roughness and vehicle transportation cost. Establishing a model to predict vehicle transportation costs based on different surface roughness conditions will be the critical feature of future work.

**Author Contributions:** Z.Z. and L.B. conceived, designed, and performed the experiments. Z.Z. analyzed the data and wrote the paper. All authors have read and agreed to the published version of the manuscript.

**Funding:** This research was supported by the Fundamental Research Funds for the Central Universities of Central South University, project number 2020zzts709.

**Acknowledgments:** The authors gratefully acknowledge the funders and all advisors and colleagues who support our work.

**Conflicts of Interest:** The authors declare no conflict of interest. The funders had no role in the design of the study; in the collection, analyses, or interpretation of data; in the writing of the manuscript, or in the decision to publish the results.

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
