# Peer review of "A New Challenge: Path Planning for Autonomous Truck of Open-Pit Mines in The Last Transport Section"

_applsci, doi:10.3390/app10186622_

Round 1
Reviewer 1 Report
- What is the novelty of the paper?
- Authors need to clearly describe the research gap in the introduction.
- Authors should provide the pseudo code for the proposed algorithm.
- Authors should clarify the problem formulation.
Reviewer 2 Report
In this paper, authors propose a modification to A* planning algorithm for mining load dump trucks. The modified algorithm takes into account the roughness of the traversable surface in addition to distance from obstacles while minimizing the overall path length from start to end position. The overall novelty of the presented work is minor, however, the paper addresses a field problem in a practical manner. The following comments are provided to the authors for improvement of the final version:
- Overall the paper is written in an understandable manner, however, there are many typos and some very hard to understand sentences e.g.
- Under equation 12 weight is misspelled as 'wight' twice.
- Line 318, it is obviously -> it is obvious
- Unnecessary usage of superlatives e.g. tremendous obstacles(line 73) -> large obstacles, outstanding roughness(Line 320) -> large roughness.
- In line 35 the word truck is missing in front in Load Haul Dump.
- Section 5, Further works should be Future Work
- Captions of images are not very meaningful e.g. Figure 3, 5, 10, 16, 17 are not descriptive enough to make them understandable on their own. Figure 15 which is the main result has a very confusing caption as there are multiple dashed and solid lines referred to with and without color.
- Images/Figures are low quality/low resolution. Legends of all images are very low resolution to understand e.g. Figure 6. Figure 9 it is not clear what the legend reads or what does the figure convey due to very low resolution. Figure 10 legends and lines are not distinguishable. Figure 15 which is the main result is very low quality and it is very hard to see the paths of the truck.
- Please double check if all abbreviations are defined before the results section as some abbreviations like OCM are never defined.
- The result of section 2.21 in Figure 3 shows many false negatives for many maxima/minima points. For example around value 18 and 65. It seem like the filtering step is not maintaining the same level of sensitivity along the dimension. Similarly XY-axes of Figure 3 should be explained more in detail as Y-axis seems to be derivative while X-axis is the vertical direction of travel?
- In 2.2.2 Xo is mentioned as Horizontal vector. I think this is incorrect as it should be vector along the line derivative is taken otherwise for vertical derivative the pkpk+1 will always result in zero dot product.
- Writing of section 2.2.3 should be improved. I understand there is no contribution here and you are adding it here for completeness, however, the overall description of the method especially around equation 5 is poor. Also the format of equation 5 is inconsistent with other equations it seems like it copy/pasted from the original paper.
- In section 2.4, C-Map is created by combining the two maps, is there a weighting applied when the two normalized maps are added together? because the maxima of two maps are different and when added together cannot result in Figure 8 results without weighted addition.
- The switch cost in equation 7 is an important element as it would penalize the cost of motion, however, no description is provided for this cost and what are its effects? It is clearly seen in all the results that the truck at least turns around once. Does this switch cost play a role?
- Please provide a table for the values of the parameters used in the experiments. What are the costs, weights max/min parameter values etc? It would also be encouraged to do studies on the effect and sensitivity of these methods on the performance of the overall algorithm.
- Only two experiments for two very similar DSM maps are provided. It would be interesting to see if the method works for different resolution of DSM maps and how much improvement it actually achieves?
- The final paragraph of section 3 and section 5 are very good in discussing the variability in performance of the parameters and its limitations. In the final version these sections should be refined and more details should be added for future researchers.
- A comment on the title of paper. "The Last Mile" does not relate to the overall path planning strategy presented in the paper or relates to improving the performance in the very last mile of the trajectory. Maybe a more appropriate title should be selected.
Round 2
Reviewer 2 Report
Authors did a good thorough revision.
Paper should be accepted automatically after minor revisions in final version.
Minor changes include:
- Improve language to make sentences simpler and more understandable.
- Figure captions can be further improved with details to make them independent understandable units.
Thank you.
